# Understanding the dimensions of sport-injury related growth: A DELPHI method approach

**Víctor J. Rubio**[1]*, **Alessadro Quartiroli**[2], **Leslie W. Podlog**[3], **Aurelio Olmedilla**[4]

**1** Department of Biological and Health Psychology, University Autonoma Madrid, Madrid, Spain,
**2** Department of Psychology, University of Wisconsin-La Crosse, La Crosse, Wisconsin, United States of America, **3** School of Medicine, University of Utah, Salt Lake City, Utah, United States of America,
**4** Department of Personality, Evaluation and Psychological Treatment, University of Murcia, Murcia, Spain

☺ These authors contributed equally to this work.
* victor.rubio@uam.es

**Data Availability Statement:** All relevant data are within the paper and its Supporting Information files.

**Funding:** The author(s) received no specific funding for this work.

## Abstract

Despite the multitude of adverse physical and psychosocial consequences of sports injury, evidence also highlights the potential for positive benefits in the injury aftermath. The purpose of this study is to address this gap by exploring the dimensions of personal growth following a severe sports injury. A 3-rounds e-Delphi method was utilized to develop a consensual understanding of the dimensions of sport-injury related growth. A panel of 24 psychology of sport injury experts participated in the process. The final list of items obtained was subjected to conventional content analysis to identify general themes. The process led to the development of a 5-dimension model capable of describing athletes' experiences of personal growth following a severe sports-related injury: personal strength, improved social life, health benefits, sport benefits, and social support and recognition. The domains of sport-injury related growth identified here are consistent with growth-domains identified in previous personal growth literature. However, they also highlight the importance of contextualizing the experience of growth. We have captured key elements of sport-injury related growth, domains that can be used as the basis for further psychometric testing and for further interventions to increase adjustment and well-being during the rehabilitation process.

## Introduction

Sport-related injuries have become a worldwide health concern [1] with incidence rates ranging from 2.4 to 4.5 per 1000 hours of training and competition exposure depending on sport modalities and levels [2]. Severe sport injuries requiring time loss from training and competition for 4 weeks or more have been shown to have deleterious implications for athletes' physical and psychosocial well-being among professional and amateur athletes alike [3,4]. Athletes have acknowledged that serious injury may be one of the most arduous emotional and physical challenges they face in their career [5].

Although ample evidence highlights the adverse consequences of sport injuries [6,7], more recent scholarship has focused on positive adaptations or growth outcomes resulting from the sport injury experience [7–9]. A range of positive adaptations have been reported, for example,

**Competing interests:** The authors have declared that no competing interests exist.

physical (e.g., strengthening weaker body parts, improved fitness), social (e.g., opportunities to engage with non-sport peers), psychological (e.g., enhanced mental toughness, greater determination to reach goals), and technical/tactical ones (e.g., improved game strategy or tactical awareness) [10–12]. In an effort to capture such benefits or adaptations, researchers have employed various terms, for example stress-related growth, post-traumatic growth, adversarial growth, and benefit finding [13].

Unfortunately, research on growth following sport-related injuries is plagued with conceptual confusion, interchangeable use of terms that may or may not be similar in nature, and a lack of clear consensus on the domains of growth. For instance, Wadey et al. argued that researchers have on occasion used the term post-traumatic growth to examine phenomena that may not be "traumatic" in nature [9]. The researchers contend that use of the term post-traumatic growth should be reserved for individuals experiencing life altering events (e.g., loss of a loved one, a near fatal accident, wartime experiences), rather than those of a lesser severity (e.g., a sport injury). Moreover, growth is usually related to changes in self-perception, changes in the experience of relationships with others, and changes in one's general philosophy of life [14]. However, given the plethora of positive adaptations or benefits associated with growth following a sport injury, the actual domains of growth remain unclear. In particular, discrepancies exist regarding the number of growth domains or benefits reported across sport injury studies, with some identifying as few as three overarching domains [7] and others reporting as many as 19 factors [9]. Finally, sport injury researchers have often utilized growth measures developed for non-sport injury populations [15, 16]. Consequently, it seems plausible that such measures fail to capture relevant components of growth specific to the sport injury context.

In order to move research in this area forward, there is an evident need to gain consensus on the key domains of growth following a sport injury and to employ consistent terminology in studying the phenomenon of interest. Consequently, the aim of the current study was to clarify existing dimensions and develop a novel, consensually agreed upon multidimensional model of personal growth following a severe (more than 4 weeks of sport absence) sport-related injury. We employ the term, sport-injury related growth, given our interest in understanding the domains of growth experienced in the aftermath of a severe (absence of 4 weeks or more from training/competition) sport-related injury. Results from this study provide an important starting point in the development of and future psychometric testing of key elements of sport-injury related growth.

## Materials and methods

The study was carried out according to Universidad Autonoma Madrid (Spain) IRB approval (UAM CEI 66–1171). Consent was obtained electronically when experts participants in the study acceded to the electronic survey.

To address our question of interest, we used a 3-rounds e-Delphi method to develop a consensual understanding of the dimensions of growth following a severe sport injury [17]. A Delphi method is a useful technique to solicit the opinions of experts in a specific domain, with the aim of reaching consensus among them [18]. This iterative process–generally occurring in three to four rounds–involves a series of ad-hoc developed questionnaires, each built on the results of the preceding questionnaire. The results of each round are compiled and returned to the experts, who over successive iterations, are able to reevaluate their responses taking in to consideration the anonymous responses of the entire panel. In the current study, we used a ranking-type Delphi design [17,18]. During the first round, we solicited the perspectives of psychology of sport injury experts who were asked to identify the salient aspects of growth

following a severe sport injury. In rounds two and three, our goal was to establish the relative importance of each of the items identified by the experts in round one.

During the first round, a panel of 40 experts were invited to participate via email in which they were informed about the study aims, why they were selected to participate and what was expected of them. The email included a link to the survey hosted in Qualtrics (Qualtrics, Provo, UT). In this study, the inclusion criteria for experts were: (a) having contributed to the current literature focused on resilience and growth issues among injured athletes within the previous 5 years, (b) having at least 5-years-experience as a scholar and/or practitioner in the field of sport psychology, (c) having listed 'psychology of injury' as one of their main areas of expertise on their website, (d) be a member of at least one sport injury interest group in professional organizations (e.g., AASP, NASPSPA, FEPSAC). The panel included professionals with an average of 14.28 (SD = 8.7) years of experience working within the psychology of injury, either as a scholar or applied sport psychology practitioner. Of the 40 leading experts who were contacted, twenty-four (USA: 12; Europe: 12) completed all three rounds of this Delphi study. These individuals included applied (7), academic (13) and applied-academic (6) professionals. Although the optimal size of a Delphi panel depends on the purpose of the study and the heterogeneity of the target population [19], empirical examination of this technique suggests that a linear increase in accuracy occurs as the number of experts in the panel exceeds 11 or more [20], with15 to 20 panelists considered optimal [21,22].

In the initial round, after giving consent, experts responded to an open-ended question ("Please list the types of benefits or positive aspects an athlete might perceive after suffering a severe injury"). Responses were collected and analyzed using an inductive content analysis procedure [23–25]. The goal of the initial, open-ended question was to solicit a diversity of opinions regarding all possible dimensions of sport-injury related growth, rather than to guide the experts towards any particular response or to achieve consensus.

Based on these results, a preliminary list of 47 items was developed to reflect possible dimensions of growth following a sport-related injury. During the second round, experts were asked to rank each item according to its appropriateness in describing a possible dimension of growth following a sport injury. Responses were recorded using a 0–5 Likert scale with 0 indicating "not at all and should be removed from the list," and 5 indicating "absolutely and should be included by all means". The data obtained from the ranking portion in round 2 were analyzed according to item homogeneity and experts' convergence, described in greater detail below. Statements lacking in homogeneity (i.e., deviation from the mean) and/or convergence (i.e., deviation from the mode) were included in the third round of the Delphi. A third round was launched sending an individualized survey to each expert, including the list of items that based on the criteria of homogeneity and convergence, still needed to reach consensus among experts. Following the Delphi method procedure, each item was introduced to the experts along with its mode value, based on the second-round experts' rating. The aim in doing so was to inform each expert of the groups' evaluation of each specific item in order to promote further reflection on the single items by individual experts. Such reflection provided an opportunity to confirm or modify one's initial evaluation of individual items.

The final list of items, developed following the Delphi methodology, was then subjected to a conventional content analysis to identify general themes [26]. Three members of the research team (authors one, two and four) independently identified themes emerging across the entire list of items through the use of inductive content analysis. This process involved an open discussion of the individuals' analysis aiming to identify a coherent and meaningful structure describing the content of the items. A final deductive analytic phase was then completed, with the same three researchers independently assigning each item to one of the 5 specific domains.

Table 1. Experts' characteristics and demographics.

| | Country of Practice | Years of Experience | Main Professional Area | Area of Specialization |
|---|---|---|---|---|
| 1 | Portugal | 21 | Applied and Academic | Sport Psychology |
| 2 | USA | 10 | Primarily Academic | Performance Psychology |
| 3 | USA | 12 | Primarily Academic | Psychology of Sport Injury & Rehabilitation |
| 4 | USA | 27 | Primarily Academic | Psychology of Sport Injury |
| 5 | USA | 7 | Primarily Applied | Sport Psychology Consulting |
| 6 | USA | 8 | Primarily Academic | Psychology of Sport Injury & Rehabilitation |
| 7 | Spain | 12 | Primarily Academic | Sport Psychology |
| 8 | Portugal | 10 | Primarily Academic | Sport Psychology and Sport Training |
| 9 | USA | 10 | Primarily Academic | Resilience and stress-related growth |
| 10 | Spain | 15 | Applied and Academic | Sport Psychology |
| 11 | Sweden | 7 | Primarily Academic | Psychology of Sport Injury |
| 12 | Spain | 10 | Applied and Academic | Sport Psychology |
| 13 | Sweden | 20 | Primarily Academic | Psychosocial perspectives of sport injury |
| 14 | Spain | 8 | Primarily Applied | Sport Psychology |
| 15 | Spain | 8 | Primarily Applied | Sport Injuries in young athletes |
| 16 | Spain | 20 | Primarily Academic | Psychology |
| 17 | USA | 15 | Primarily Academic | Psychology of Sport Injury & Rehabilitation |
| 18 | USA | 19 | Primarily Applied | Psychology of High Performance |
| 19 | USA | 13 | Primarily Academic | Sport Psychology |
| 20 | USA | 10 | Applied and Academic | Mental Performance Optimization |
| 21 | France | 6 | Applied and Academic | Psychology of Sport Injury & Rehabilitation |
| 22 | Sweden | 31 | Applied and Academic | Sport Psychology |
| 23 | USA | 9 | Primarily Academic | Sport Psychology |
| 24 | USA | 30 | Primarily Academic | Sport Psychology |
| | Mean | 14.08 | | |
| | SD | 7.36 | | |

This phase was completed through group discussion, the aim being to establish agreement regarding the extent to which individual items reflected the larger themes.

## Results

Findings from the current study were the product of a three-phase data collection and analysis process. Of the 40 experts initially invited for study participation, 24 participated in all three rounds (response rate = 60%) (Table 1).

### Round 1

Of the 40 experts invited to participate in the study, 30 initially participated in the first round of the study (response rate 75%). These experts provided a total of 173 responses in the form of short narrative and single statements describing personal growth-related aspects of sport injury. After receiving the panelists' answers to the open-ended question, an inductive content analysis was completed [2,7,25]. A list of 47 unique statements (i.e., items) emerged which were subsequently used in the second Delphi round.

### Round 2

In round 2, the finalized list of 47 items was sent to all 30 experts, however only 24 responded, partially reducing the sample size of the study (80% Response rate). Their responses were

analyzed with respect to the level of homogeneity and convergence reached for each item. In terms of homogeneity, each item presenting a standard deviation to the mean greater or equal to 1, was considered lacking experts' consensus, therefore lacking homogeneity [27]. On the other hand, when an expert's rating was above/below the item's Mode ±1, the specific item was considered lacking in convergence. Based on this analysis of the data, all statements had a mode of 3 or greater and were therefore retained. One statement did not reach the mode of 3 and was eliminated from further analysis. While fourteen of the total 46 items, reached both homogeneity and convergence, the remaining 32 items showed a standard deviation to the mean greater than or equal to 1, and for this reason, the 32 items were selected to be included in the third round.

## Round 3

In round 3, following suggestions by Keeney et al., each of the 24 panelists received a link to a survey presenting back to them the 32 items for which the panelists' ratings differed by two or more levels from the item's group Mode. In the end, a finalized list of 46 items, upon which the experts reached consensus (within ±1 Mode), was completed [27]. Among these items were the 14 items that reached consensus in the second round and the remaining 32 items that reached consensus during the third round. As part of the inductive analysis, all of the single items were grouped into sequentially more abstract themes reflecting the key dimensions of sport-injury related growth. Based on the conceptual underpinnings of the 46 statements, the five dimensions were labeled as: personal strength, improved social life, health-related benefits, sport-related benefits, and social support and recognition (see Table 2). These domains were defined as follows:

### Personal strength

Improving and fostering personal, non-sport-related skills and competencies such as resiliency and coping skills, increased empathy, and trust in others.

### Improved social life

Enhanced and improved social life in the form of increased time spent with friends or relatives and the possibility of strengthening current relationships and developing new ones. Moreover, this dimension represents an increased appreciation for the importance of significant others.

### Health-related benefits

Learning about how the body functions, injury mechanisms and treatments, and the role of health-related processes such as nutrition and rest. This dimension also included athlete's learning about injury prevention measures and greater awareness of sport-related/professional support systems (e.g., sport medicine services, physical therapists).

### Sport-related benefits

Taking advantage of the time out of athletic competition to increase sports skills and/or physical fitness (e.g. having the opportunity to work on fitness, strength, endurance, and flexibility) and to promote the exploration and learning of other sport-related functions (e.g., officiating, coaching). Moreover, this dimension included athletes' ability to actively assume different roles to support teammates or coaches (e.g., stats keeping).

**Table 2. Final items reflecting dimensions of sport-injury related growth.**

| Personal Strength |
|---|
| I have become a more empathic person |
| I have transcended my own limits by rising above adversity |
| I have re-defined my identity beyond sport |
| I have understood that sport-related social recognition is transitory |
| I have developed mental toughness |
| I have found the time to develop non-sport related interests |
| I have become aware of my limits |
| I have challenged myself, both mentally and physically |
| I have learned to appreciate the small things in life |
| I have enhanced my spiritual strength |
| I have had the opportunity to think about what is important in my life |
| I have learned to interpret my injury as a sign that it's time to move on from sport before something more severe happens |
| **Improved Social Life** |
| I have a greater understanding of the importance of the people close to me |
| I have had time to foster personal relationships, including those outside of sport (e.g., family, friends) |
| I have had time to socialize and hang out with friends outside of my sport |
| I have learned to trust others more than before |
| My relationships with teammates and/or coaches have improved |
| My awareness of belonging to my group/team has grown |
| I have developed relationships with other (former or currently) injured athletes |
| I have learned to appreciate other people's contributions inside and outside of sport |
| I have realized who I can really count on |
| **Health-related Benefits** |
| My appreciation for medical and healthcare professionals has deepened |
| I have identified support systems and resources that I can now rely on |
| My ability to manage physical pain in the future has improved |
| I have learned to avoid high-risk behaviors and/or to develop injury prevention measures |
| I have become more appreciative of maintaining good health and living free from injury |
| I have broadened my knowledge of my injury, my body, and/or health guidelines (e.g., nutrition, rest) |
| My body has had the chance to recover from training and competition |
| **Sport-related Benefits** |
| I have become a role model for other athletes (e.g., peers, young people) |
| I have learned to act as referee or coach |
| I have engaged in different roles supporting my team (e.g., stats keeping) |
| I have found a more valuable role within my team |
| I have gained a new perspective on sport |
| I have learned to appreciate opportunities for playing sports rather than taking them for granted |
| I have developed mental skills that are applicable to sports |
| I have been able to work on fitness, strength, endurance, and flexibility |
| I have had the opportunity to work on technical aspects of sport, strategy and/or tactical awareness |
| I have improved my ability to handle pressure and/or to deal with competitive stress |
| I have furthered my commitment, motivation, and passion for sport |
| My ability to differentiate between pain from practicing sport and injury-related pain has improved |
| **Social Support and Recognition** |
| I have seen that others value me as a person and not just as an athlete |
| I have received support for daily needs and responsibilities |

*(Continued)*

**Table 2.** (Continued)

| |
| --- |
| I have received encouragement and/or emotional support |
| I have received attention and/or care from significant others |
| I have earned recognition and praise for my efforts during my rehabilitation |
| I have earned the respect of others for having sacrificed my body for sport |

## Social support and recognition

Increased appreciation of the emotional support received from friends, teammates, and coaches in order to successfully cope with injury and daily life challenges during the injury time-frame. This dimension also refers to increased respect and social recognition garnered as a consequence of athletes' ability to endure arduous rehabilitation regimens in order to participate in one's chosen sport.

## Discussion

Severe sport injuries can threaten athletes' sporting objectives and have been shown to have a range of detrimental impacts on athletes' personal, social, academic and intra-psychic functioning [3,6]. As such, sport injuries represent a clear form of adversity [28]. Despite the detrimental implications of injury, a growing body of empirical research, has in recent years, uncovered a range of positive consequences emanating from the sport injury experience [29]. Until now however, research in this area has been hampered by a lack of consensus regarding the domains of personal growth following severe sport-related injuries.

Using a knowledge-driven consensual Delphi Method, we sought to identify the conceptual domains of sport-injury related growth. A panel of 24 sport psychology researchers and practitioners, offered insights into relevant domains of growth following injury. Expert responses support previous research suggesting the possibility that positive outcomes may ensue in the aftermath of seemingly adverse events [13–16]. Using a three-round Delphi technique, experts reached consensus on a list of 46 statements which were subsequently content analyzed and grouped into five domains (see Table 2): personal strength, improved social life, health-related benefits, sport-related benefits, and social support and recognition.

The five domains emerging from the current study are consistent with growth-domains identified in previous health and positive psychology research. For example, personal strengths and improved social life identified in the present investigation are consistent with changes in self-perception, changes in the experience of relationships with others, and changes in one's general philosophy of life [14]. Despite such similarities, the domains identified by experts in our study also highlight the importance of contextualizing the experience of adversity [13,15]. Thus, they also included sport- and relevant domains, such as health-related benefits, sport-related benefits, as well as social support and recognition. Findings from the current study also appear instrumental in synthesizing the diverse array of findings regarding the dimensions of growth following a sport-related injury. Results provide the basis for further psychometric testing of a much-needed sport-injury specific measure of growth. Such a measure would be invaluable in facilitating examination of growth antecedents and outcomes in a sport injury context. The domains identified can also be used to further explore antecedents and outcomes of growth, including, but not limited to rehabilitation processes, return to play, and drop-out processes, and to put the light of psychologists working in sport medicine in promoting the positive aspects to improve such processes.

While the study has important theoretical and practical implications, several limitations are evident. First, results that emerge using a Delphi method necessarily rely on the subjective

experiences, values, and beliefs of the experts participating in the investigation. Although we attempted to solicit a robust sample size, the experts chosen were not randomly selected. As such, it is possible that the cultural and professional background of the participating experts may have influenced the results. Second, even though the number of experts who participated in the study exceeded Delphi guidelines, the fact that only 24 of the 40 experts contacted participated in the study, indicates we might have missed relevant insights into our question of interest. Third, it is possible that inclusion of a broader range of experts, for example sport medicine professionals such as physiotherapists, surgeons, or sport medicine physicians might have elicited novel insights into the dimensions of sport-injury related growth. Fourth and finally, in the present study, we did not examine whether athletes actually or objectively achieved sport-injury related growth. Further research is needed to match perceptions of sport-injury related growth with behavioral or objective indicators of growth. Despite these limitations, the present study advances the literature by providing a more definitive understanding of the dimensions of growth following a sport-related injury. Such understanding will facilitate development of and future psychometric testing of key elements of sport-injury related growth.

## Supporting information

**S1 Dataset. Delphi survey personal growth in injury—1st round.**
(XLSX)

## Acknowledgments

Authors wish to thank the experts who have participated in the study. We also would like to thank José Manuel Hernández and María Oliva Márquez for their suggestions in designing the Delphi study.

## Author Contributions

**Conceptualization:** Víctor J. Rubio.

**Data curation:** Víctor J. Rubio, Alessadro Quartiroli.

**Formal analysis:** Víctor J. Rubio, Alessadro Quartiroli.

**Investigation:** Aurelio Olmedilla.

**Methodology:** Víctor J. Rubio, Alessadro Quartiroli.

**Project administration:** Víctor J. Rubio.

**Resources:** Aurelio Olmedilla.

**Supervision:** Víctor J. Rubio.

**Writing – original draft:** Víctor J. Rubio, Alessadro Quartiroli, Leslie W. Podlog.

**Writing – review & editing:** Víctor J. Rubio, Alessadro Quartiroli, Leslie W. Podlog, Aurelio Olmedilla.

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
