## [Decision Letter · Decision Letter 0]

30 Mar 2020

PONE-D-20-00634

Understanding the dimensions of sport-injury related growth: A DELPHI method approach

PLOS ONE

Dear Dr Rubio,

Thank you for submitting your manuscript to PLOS ONE. After careful consideration, we feel that it has merit but does not fully meet PLOS ONE’s publication criteria as it currently stands. Therefore, we invite you to submit a revised version of the manuscript that addresses the points raised during the review process.

We would appreciate receiving your revised manuscript by May 14 2020 11:59PM. To enhance the reproducibility of your results, we recommend that if applicable you deposit your laboratory protocols in protocols.io, where a protocol can be assigned its own identifier (DOI) such that it can be cited independently in the future. For instructions see: http://journals.plos.org/plosone/s/submission-guidelines#loc-laboratory-protocols

We look forward to receiving your revised manuscript.

Kind regards,

Geilson Lima Santana, M.D., Ph.D.

Academic Editor

PLOS ONE

Journal Requirements:

2. Please provide additional details regarding participant consent. In the ethics statement in the Methods and online submission information, please ensure that you have specified whether consent was informed.

Reviewers' comments:

Reviewer's Responses to Questions

**Comments to the Author**

1. Is the manuscript technically sound, and do the data support the conclusions?

Reviewer #1: Partly

2. Has the statistical analysis been performed appropriately and rigorously? 

Reviewer #1: N/A

3. Have the authors made all data underlying the findings in their manuscript fully available?

Reviewer #1: No

4. Is the manuscript presented in an intelligible fashion and written in standard English?

Reviewer #1: No

5. Review Comments to the Author

Reviewer #1: The topic of the study is not only relevant, as descried by the authors, but also very interesting!

In the introduction the authors have set the stage, explaining why the study matters and putting the research in context by identifying other literature on the topic. Well done. In the brief literature review on the topic was not clear if the athletes in questions are always and only professional athletes.

The study aim is stated in line 52, but it was not sufficiently clear if the aim of the study was to identify new domains or clarify existing dimensions of personal growth following a severe sport-related injury. Also, the study objectives were lacking. Moreover, it was not clear if authors sought to measure the diversity of opinions on the topic or to steer the group towards consensus.

I believe the quality of the paper would improve if authors clearly stated why the Delphi method was chosen and why it was the most appropriate method to answer their research question.

It would be important to clearly state and outline the panellist inclusion criteria as means of evaluating the results and stablishing the study’s potential relevance to other settings and populations.

The authors have clearly stated how ‘expertise’ of panellists was define and conceptualized in their study (line 75) and have included a diverse and varied panel (line 74). Well done.

To make the study methodologically robust, a description of any pilot study of rounds 1 and 2 should be included, as well as how the round 1 open-ended question was formulated.

In the results section, it wasn’t clear how many panellists contributed to all 3 rounds (discrepancy between line 72 and line 114).

The authors have clearly described the rounds and the analysis conducted on each stage. However, reporting results on each round separately would illustrate clearly the array of themes generated in round one and give an indication of the strength of support for each round. The presentations of findings are important and findings from subsequent rounds should be reported in a summarized format to indicate the relative standing of each of the opinions (e.g. how many changes have been made after round 3?).

The tables in the paper are clear and readable, the presentation is appropriate for the information being presented, supporting the study findings. I would suggest amending Table 2 caption to highlight its information.

In the discussion section, the results support author’s conclusions, however authors do not discuss any limitations of their study. Also, it was not clear how the identified domains ‘emphasize the importance of interaction between the stressor faced by the individual and the context in which the stressor is experienced’ (lines 195-197).

I would also suggest proofreading the paper for use of English language and typing mistakes (e.g. line 64 ‘The Study During the first round’).

6. PLOS authors have the option to publish the peer review history of their article (what does this mean?). If published, this will include your full peer review and any attached files.

Reviewer #1: Yes: Paula de Vries Albertin

---

## [Author Response · Author response to Decision Letter 0]

15 May 2020

Journal requirements

1. Please ensure that your manuscript meets PLOS ONE's style requirements, including those for ﬁle naming. The PLOS ONE style templates can be found at http://www.plosone.org/attachments/PLOSOne_formatting_sample_main_body.pdf and http://www.plosone.org/attachments/PLOSOne_formatting_sample_title_authors_afﬁliations.pdf

- We have revised our manuscript to meet all PLOS ONE’s style requirements.

2. Please provide additional details regarding participant consent. In the ethics statement in the Methods and online submission information, please ensure that you have speciﬁed whether consent was informed.

- Experts’ consent was given when they accessed the first round of the Delphi method. Unless they clicked on the “Ok button”, they could not complete the questionnaire. Moreover, participants initially received an email in which they were informed about the study, how they were selected and what was expected of them. We have included mention of this point in the revised Method section (lines 82-84). The IRB of the first author’s institution approved this procedure.

We have also included the suggested specific wording at the beginning of the Methods section: "Universidad Autonoma Madrid (Spain) IRB approval (UAM CEI 66-1171).

Consent was obtained electronically when experts participants in the study acceded to

the electronic survey"

3. We note that you have indicated that data from this study are available upon request. PLOS only allows data to be available upon request if there are legal or ethical restrictions on sharing data publicly. For more information on unacceptable data access restrictions, please see http://journals.plos.org/plosone/s/data-availability#loc-unacceptable-data-access-restrictions. In your revised cover letter, please address the following prompts:

a) If there are ethical or legal restrictions on sharing a de-identiﬁed data set, please explain them in detail (e.g., data contain potentially sensitive information, data are owned by a third-party organization, etc.) and who has imposed them (e.g., an ethics committee). Please also provide contact information for a data access committee, ethics committee, or other institutional body to which data requests may be sent.

b) If there are no restrictions, please upload the minimal anonymized data set necessary to replicate your study ﬁndings as either Supporting Information ﬁles or to a stable, public repository and provide us with the relevant URLs, DOIs, or accession numbers. For a list of acceptable repositories, please see http://journals.plos.org/plosone/s/data-availability#loc-recommended-repositories.

- We are uploading as Supporting Information a compressed excel file including the DELPHI 3-round dataset.. 

Answers to the Reviewer:

- In the introduction the authors have set the stage, explaining why the study matters and putting the research in context by identifying other literature on the topic. Well done. In the brief literature review on the topic was not clear if the athletes in questions are always and only professional athletes.

As requested, we have clarified the fact that research in this area has included a variety of samples, not just professional athletes. In fact, most of the cited articles used amateur instead of professional athletes. Along these lines, we have added a sentence clarifying this point on lines 26-27.

- The study aim is stated in line 52, but it was not sufficiently clear if the aim of the study was to identify new domains or clarify existing dimensions of personal growth following a severe sport-related injury. 

Also, the study objectives were lacking. 

Moreover, it was not clear if authors sought to measure the diversity of opinions on the topic or to steer the group towards consensus.

The reviewer raises several important points relative to our study aim. As suggested, we have indicated in the revised version that our goal was to clarify existing dimensions. Also, as indicated in the initial submission on line 50, our ultimate aim was to “gain consensus” on the key domains of growth. Moreover, the nature of the Delphi method is to build consensus among the panelists. That said, experts participating in the study were not in any way “steered towards” consensus; as indicated in our initial description of the methods (line 101), in the first round of the Delphi method, we asked experts a broad, open-ended question, specifically to “list the types of benefits or positive aspects an athlete might perceive after suffering a severe injury.” In line with the reviewer’s comment, we have added the sentence on lines 58-61: “Consequently, the aim of the current study was to “clarify existing dimensions and develop a novel consensually agreed multidimensional model of personal growth following a severe (more than 4 weeks of sport absence) sport-related injury.” 

We have also added in a statement on lines 103-106, where we indicate “The goal of the initial, open-ended question was to solicit a diversity of opinions regarding all possible dimensions of sport-injury related growth, rather than to guide the experts towards any particular response or to achieve consensus”. We hope the aforementioned updates help clarify the study objectives.

- I believe the quality of the paper would improve if authors clearly stated why the Delphi method was chosen and why it was the most appropriate method to answer their research question.

Thank you for the suggestions! We have added a paragraph that we believe addresses the reviewer’s comment on ln 71-81 where we indicate: “A Delphi method is a useful technique to solicit the opinions of experts in a specific domain, with the aim of reaching consensus among them [20]. This iterative process–generally occurring in three to four rounds–involves a series of ad-hoc developed questionnaires, each built on the results of the preceding questionnaire. The results of each round are compiled and returned to the experts, who over successive iterations, are able to reevaluate their responses taking in to consideration the anonymous responses of the entire panel. In the current study, we used a ranking-type Delphi design [17,18]. During the first round, we solicited the perspectives of psychology of sport injury experts who were asked to identify the salient aspects of growth following a severe sport injury. In rounds two and three, our goal was to establish the relative importance of each of the items identified by the experts in round one.”

- It would be important to clearly state and outline the panellist inclusion criteria as means of evaluating the results and stablishing the study’s potential relevance to other settings and populations.

We have addressed the reviewer’s point on lines 85- 91 of the revised manuscript. It now reads: “In this study, the inclusion criteria for experts were: (a) having contributed to the current literature focused on resilience and growth issues among injured athletes within the previous 5 years, (b) having at least 5-years-experience as a scholar and/or practitioner in the field of sport psychology, (c) having listed ‘psychology of injury’ as one of their main areas of expertise on their website, (d) be a member of at least one sport injury interest group in professional organizations (e.g., AASP, NASPSPA, FEPSAC).”

- The authors have clearly stated how ‘expertise’ of panellists was define and conceptualized in their study (line 75) and have included a diverse and varied panel (line 74). Well done.

Thank you for the feedback!

- To make the study methodologically robust, a description of any pilot study of rounds 1 and 2 should be included, as well as how the round 1 open-ended question was formulated.

The reviewer raises an interesting point. As iteration is part and parcel of the Delphi method, pilot work is essentially built into the design of the Delphi method. That is, since rounds 1 and 2 are phases of the research design used in a Delphi study, no pilot study was completed for in the current investigation. 

The round 1 open-ended question was formulated based on the overall aim of the study (‘Consequently, the aim of the current study was to clarify existing dimensions and develop a novel, consensually agreed upon multidimensional model of personal growth following a severe (more than 4 weeks of sport absence) sport-related injury.”, lines 59-61). Based on their expertise, experts were asked to provide insights regarding any positive aspects of the sport injury experience. 

- In the results section, it wasn’t clear how many panellists contributed to all 3 rounds (discrepancy between line 72 and line 114).

Thank you for noticing the mistake. We have corrected this issue on lines 93-94, where we indicate that: “Of the 40 leading experts who were contacted, twenty-four (USA: 12; Europe: 12) completed all three rounds of this Delphi study.”

- The authors have clearly described the rounds and the analysis conducted on each stage. However, reporting results on each round separately would illustrate clearly the array of themes generated in round one and give an indication of the strength of support for each round. The presentations of findings are important and findings from subsequent rounds should be reported in a summarized format to indicate the relative standing of each of the opinions (e.g. how many changes have been made after round 3?).

Thank you for the recommendation. As suggested, we have expanded and restructured the results section on lines 141 – 150.

- The tables in the paper are clear and readable, the presentation is appropriate for the information being presented, supporting the study findings.

Thank you for the positive feedback!

- I would suggest amending Table 2 caption to highlight its information.

Thank you for the positive comments regarding the tables. As suggested, we have amended Table 2 caption to: “Final items reflecting dimensions of sport-injury related growth” 

- In the discussion section, the results support author’s conclusions, however authors do not discuss any limitations of their study. Also, it was not clear how the identified domains ‘emphasize the importance of interaction between the stressor faced by the individual and the context in which the stressor is experienced’ (lines 195-197).

Consistent with the reviewer’s suggestion, we have highlighted the limitations of the study on lines 231-249. Moreover, we have added several sentences in the Introduction section where we emphasize the role of the interaction between the individual and the context in which the stressor is experienced (lines 45-55).

- I would also suggest proofreading the paper for use of English language and typing mistakes (e.g. line 64 ‘The Study During the first round’).

We note, the manuscript was proofread and edited by a native English speaker prior to the initial submission. Nevertheless, as suggested, this individual proofread the revised submission to ensure proper English language throughout. We believe the paper is free from any grammatical, punctuation, or sentence structure issues. Example changes include: 

1. we deleted “The Study” on line 85; 

2. we replaced “of” with “in”, in the statement on lines 112-113 “The data obtained from the ranking portion of round 2” to “The data obtained from the ranking portion ‘in’ round 2”; 

3. we replaced “researcher team” with “research team” on line 126; 

4. We indented the paragraph on line 214 with the sentence beginning “The five domains emerging from the current study are consistent…”;

5. Line 192 an “ s’ ” has been added to “ athletes’” after “of” and before “ability.”

---

## [Decision Letter · Decision Letter 1]

10 Jun 2020

Understanding the dimensions of sport-injury related growth: A DELPHI method approach

PONE-D-20-00634R1

Dear Dr. Rubio,

We’re pleased to inform you that your manuscript has been judged scientifically suitable for publication and will be formally accepted for publication once it meets all outstanding technical requirements.

Kind regards,

Geilson Lima Santana, M.D., Ph.D.

Academic Editor

PLOS ONE

Additional Editor Comments (optional):

Reviewers' comments:

Reviewer's Responses to Questions

**Comments to the Author**

1. If the authors have adequately addressed your comments raised in a previous round of review and you feel that this manuscript is now acceptable for publication, you may indicate that here to bypass the “Comments to the Author” section, enter your conflict of interest statement in the “Confidential to Editor” section, and submit your "Accept" recommendation.

Reviewer #1: All comments have been addressed

2. Is the manuscript technically sound, and do the data support the conclusions?

Reviewer #1: (No Response)

3. Has the statistical analysis been performed appropriately and rigorously? 

Reviewer #1: N/A

4. Have the authors made all data underlying the findings in their manuscript fully available?

Reviewer #1: (No Response)

5. Is the manuscript presented in an intelligible fashion and written in standard English?

Reviewer #1: Yes

6. Review Comments to the Author

Reviewer #1: Thank you for submitting a revised version of the paper. The quality of the paper has significantly improved. Well done!

7. PLOS authors have the option to publish the peer review history of their article (what does this mean?). If published, this will include your full peer review and any attached files.

Reviewer #1: Yes: Paula de Vries Albertin

---

## [Editor Report · Acceptance letter]

12 Jun 2020

PONE-D-20-00634R1 

Understanding the dimensions of sport-injury related growth: A DELPHI method approach 

Dear Dr. Rubio:

I'm pleased to inform you that your manuscript has been deemed suitable for publication in PLOS ONE. Congratulations! Your manuscript is now with our production department. 

Kind regards, 

on behalf of

Dr. Geilson Lima Santana 

Academic Editor

PLOS ONE